# Investigating COVID-19 Vaccine Impact on the Risk of Hospitalisation through the Analysis of National Surveillance Data Collected in Belgium

**DOI:** 10.3390/v14061315

**Published:** 2022-06-16

**Authors:** Diana Erazo, Maria F. Vincenti-Gonzalez, Joris A. F. van Loenhout, Pierre Hubin, Mathil Vandromme, Piet Maes, Maxime Taquet, Johan Van Weyenbergh, Lucy Catteau, Simon Dellicour

**Affiliations:** 1Spatial Epidemiology Lab (SpELL), Université Libre de Bruxelles, 1050 Brussels, Belgium; 2Scientific Directorate of Epidemiology and Public Health, Sciensano, 1050 Brussels, Belgium; joris.vanloenhout@sciensano.be (J.A.F.v.L.); pierre.hubin@sciensano.be (P.H.); mathil.vandromme@sciensano.be (M.V.); 3Laboratory of Clinical and Epidemiological Virology, Rega Institute for Medical Research, Department of Microbiology, Immunology and Transplantation, KU Leuven, 3000 Leuven, Belgium; piet.maes@kuleuven.be (P.M.); johan.vanweyenbergh@kuleuven.be (J.V.W.); 4Department of Psychiatry, University of Oxford, Oxford OX1 2JD, UK; maxime.taquet@psych.ox.ac.uk

**Keywords:** COVID-19, SARS-CoV-2, vaccination impact, risk ratio, hospitalisation surveillance

## Abstract

The national vaccination campaign against SARS-CoV-2 started in January 2021 in Belgium. In the present study, we aimed to use national hospitalisation surveillance data to investigate the recent evolution of vaccine impact on the risk of COVID-19 hospitalisation. We analysed aggregated data from 27,608 COVID-19 patients hospitalised between October 2021 and February 2022, stratified by age category and vaccination status. For each period, vaccination status, and age group, we estimated risk ratios (RR) corresponding to the ratio between the probability of being hospitalised following SARS-CoV-2 infection if belonging to the vaccinated population and the same probability if belonging to the unvaccinated population. In October 2021, a relatively high RR was estimated for vaccinated people > 75 years old, possibly reflecting waning immunity within this group, which was vaccinated early in 2021 and invited to receive the booster vaccination at that time. In January 2022, a RR increase was observed in all age categories coinciding with the dominance of the Omicron variant. Despite the absence of control for factors like comorbidities, previous infections, or time since the last administered vaccine, we showed that such real-time aggregated data make it possible to approximate trends in vaccine impact over time.

## 1. Introduction

The emergence of novel severe acute respiratory syndrome coronavirus 2 (SARS-CoV-2), which causes coronavirus disease 2019 (COVID-19), has now resulted in more than 480 million confirmed cases and 6 million reported deaths globally (30 March 2022; www.who.int). Since the start of the pandemic, governments around the world have introduced non-pharmaceutical interventions to control the spread of the virus and reduce its effective reproduction number [1]. Likewise, therapeutic management of hospitalised adults with COVID-19 is continually improving, with better clinical outcomes and recovery rates [2,3,4,5,6]. However, one fundamental phase in mitigating the COVID-19 pandemic consisted of the development and deployment of effective COVID-19 vaccines. In many countries, vaccination campaigns started in December 2020, first targeting individuals at higher risk, while gradually administering vaccines to the rest of the population. Nowadays, 35 COVID-19 vaccines have been approved by at least one country, and 11.2 billion doses have been administered across 184 countries [7].

In Belgium, the nationwide vaccination campaign against COVID-19 was launched on 5 January 2021, shortly after an 8-day pilot phase in three nursing homes. The campaign was organised in successive phases, first targeting nursing home staff and residents and then healthcare staff in hospitals. These were followed by healthcare workers in primary healthcare settings, residents of collective care institutions, people aged 65 years and older, and persons aged 18–64 years with comorbidities associated with increased risk of severe COVID-19. The vaccine campaign was then extended to the general population aged 18 years and older, before opening to those aged 12 years and older in summer 2021, and to those aged 5–11 years old from December 2021 onwards. Since September 2021, a booster dose is recommended in Belgium and has been progressively administered to the population, mostly following the order of prioritisation used during the initial vaccination campaign. To date, a total of five vaccines against COVID-19 have been approved for use by the Belgian authorities, namely BNT162b2 (Pfizer), mRNA-1273 (Moderna), ChAdOx1 (AstraZeneca), Ad26.COV2.S (Johnson & Johnson), and more recently NVX-CoV2373 (Novavax). On 30 March 2022, the majority of people who had received their primary course were administered the BNT162b2 vaccine (72%), followed by ChAdOx1 (15%), mRNA-1273 (8%), and Ad26.COV2.S (5%). Booster doses have almost exclusively consisted of mRNA vaccines (BNT162b2, mRNA-1273), and as of 28 February 2022, no NVX-CoV2373 vaccine had been distributed. As of 30 March 2022, 79% of the total Belgian population has been fully vaccinated and 62% have received one booster dose (epistat.wiv-isp.be/COVID, accessed on 10 March 2022).

Despite the proven high efficacy of the vaccines in controlling hospitalisation and deaths associated with SARS-CoV-2 infection, the emergence and circulation of variants of concern (VOCs) with novel antigenic profiles has been found to reduce the vaccine effectiveness [8,9,10,11]. In Belgium, this was the case for the Delta variant and even more for the Omicron variant, which started circulating on the Belgian territory around May and November 2021, respectively [12].

Evaluating the COVID-19 vaccine impact on the risk of hospitalisation is crucial when aiming to monitor the waning of immunity conferred by COVID-19 vaccines and related level of protection against severe illness within the population over time, as well as the potential impact of new emerging SARS-CoV-2 variants associated with relatively higher immune escape [13,14,15]. With this aim, we used hospitalisation data from the Surge Capacity (SC) survey, a survey developed by Sciensano, the Belgian Scientific Institute of Public Health. The SC survey is a mandatory survey, collecting exhaustive daily aggregated data on the number of hospitalised COVID-19 patients in Belgian general hospitals [16]. Since 6 October 2021, new variables have been added to this survey to allow hospitals to report the number of hospitalised patients by vaccination status in different age categories.

The overall goal of the present study is to approximate the recent evolution of COVID-19 vaccine impact through risk ratios estimated from real-time hospital surveillance data. Such aggregated surveillance data differ from individual data that could be collected and linked from several existing registries as it is done, for instance, in the LINK-VACC project, allowing post-authorisation surveillance to be undertaken of the COVID-19 vaccines in Belgium [17]. Indeed, while aggregated hospitalisation surveillance facilitates the acquisition of larger and more representative data sets, e.g., collected on a national scale, it does not necessarily lend itself to controlling for external factors (such as potential comorbidities or duration since the last administered vaccine dose) as a prospective cohort study would [18,19]. However, working on such aggregated surveillance data has a couple of advantages. First, General Data Protection Regulation (GDPR) issues do not apply, unlike data sets consisting of individual data. Second, the associated surveillance network is easier to put into place and makes it possible to collect highly representative data in near-real-time. The objective of the present study is to explore the utility of these aggregated surveillance data for tracking potential trends in the impact of vaccines relative to the risk of hospitalisation. Specifically, we aim to leverage this exhaustive Belgian surveillance data set to evaluate the evolution of reduced risk of hospitalisation of the vaccinated compared to the unvaccinated portion of the population across different age categories.

## 2. Data Collection

The data analysed in this study were obtained from two distinct sources: (i) non-publicly available hospitalisation surveillance data from the SC survey collected and curated by Sciensano (hereafter referred to as the SC survey data set), and (ii) publicly available vaccination data at the Belgian population-level retrieved from the Sciensano website (epistat.wiv-isp.be/COVID, accessed on 10 March 2022; hereafter referred to as the vaccine coverage data set). In addition, the number of Belgian people in each age category in 2021 was obtained from the StatBel database (bestat.statbel.fgov.be, accessed on 10 March 2022). SC survey data were collected from 6 October 2021 to 28 February 2022, and correspond to the daily number of new hospitalisations categorised as being due to COVID-19, aggregated by age category (0–11, 12–17, 18–24, 25–34, 35–44, 45–54, 55–64, 65–74, 75–84, and 85 years old and over, hereafter referred to as “85+”), and by vaccination status (unvaccinated, partially vaccinated, primary course completed, booster administered*, and unknown vaccination status). (*) This category encompasses people having received two doses of BNT162b2, mRNA-1273, or ChAdOx1 plus one booster shot, but also one shot of the Ad26.COV2.S vaccine plus one booster shot. Furthermore, in the SC survey, hospitalised patients having had their booster jab could only be registered as “boosted” from 8 December 2021 onwards. Before that date, these patients were registered as “primary course completed”. Therefore, in this analysis, we were only able to make the distinction between the complete primary course and booster vaccination schemes for the second half of the study period (15/12/21–28/02/22). The vaccine coverage data set included vaccines administered by date, region, age, sex, brand, and dose (“A”: one out of two doses, “B”: two out of two doses, “C”: one out of one Ad26.COV2.S dose, and “E”: booster dose) from 28 December 2020, to 28 February 2022. See Appendix A for a summary of the correspondence between each vaccination status and the vaccination category considered in the SC survey and vaccine coverage data sets.

## 3. Data Analyses

We estimated a risk ratio (RR) as the ratio of risk of hospitalisations between two subgroups of the population (e.g., “primary course completed” vs. unvaccinated, or “boosted” vs. “non-boosted”). For one subgroup, the risk of hospitalisation was estimated as *n*_hosp_/*n*_pop_, where *n*_hosp_ is the number of COVID-19 hospital admissions for that subgroup and *n*_pop_ is the total size of that subgroup. For instance, the RR value corresponding to the ratio between the risk of hospitalisation for the population with a primary course completed and the risk of hospitalisation for the unvaccinated population was estimated as follows:$$\begin{array}{ll} \mathrm{RR} & =\left(p_{hospitalised|vaccinated}\right)/\left(p_{hospitalised|unvaccinated}\right)\\ & =\left(n_{hospitalised,vaccinated}/n_{vaccinated}\right)/\left(n_{hospitalised,unvaccinated}/n_{unvaccinated}\right) \end{array}$$
where $n_{hospitalised,vaccinated}$ and $n_{hospitalised,unvaccinated}$ are the number of COVID-19 hospital admissions of vaccinated (primary course completed in the example) and unvaccinated patients (in a given period of time and age category), respectively, and $n_{vaccinated}$ (primary course) and $n_{unvaccinated}$ are the number of vaccinated and unvaccinated people, respectively, in the corresponding population and time period.

We estimated RR values for different periods and vaccination schemes (Appendix A), as well as for each age category involving a sufficiently large hospitalisation incidence (i.e., >1000 hospital admissions associated with a known vaccination status during the study period), i.e., all age categories from 25 years onwards. Specifically, we performed two different kinds of analyses: analyses performed on fixed periods, and analyses based on a retrospective sliding window. First, we performed six distinct analyses based on a fixed period: an analysis based on (i) the first half of the study period (06/10/21–14/12/21) for the RR associated with “at least primary course completed” (primary course completed, boosted or not) vs. unvaccinated, (ii) the same as (i) but for the second half of the study period (15/12/21–28/02/22), (iii) the second half of the study period (15/12/21–28/02/22) for “only primary course” (strictly 2/2 or 1/1 dose) vs. unvaccinated, (iv) boosted vs. unvaccinated and (v) boosted vs. “only primary course completed”, and (vi) the entire study period (06/10/21–28/02/22) for “at least primary course completed” vs. unvaccinated. Second, we performed sliding window analyses based on a retrospective time window of four weeks: for each successive day *d*, we estimated a RR value based on the hospitalisation data available between *d* - 28 days and *d*. In addition to RR estimates, we also computed and reported associated 95% confidence intervals as described by Morris & Gardner [20].

For all the different periods defined above, i.e., the fixed periods or the successive 4-week periods for the sliding windows, we considered a lag time of two weeks between the vaccination and hospitalisation data; for a specific period, we considered the vaccination coverage two weeks before the midpoint of this period. The rationale behind this choice is that considering a lag of two weeks before the starting day of each period would lead to a potential overestimation of the RR because we would underestimate the vaccination coverage when getting closer to the end of that period. Conversely, considering a lag of two weeks before the ending day of each period would lead to a potential underestimation of the RR because we would overestimate the vaccination coverage at the beginning of that period. Therefore, considering the midpoint of each period minus the lag time appears as a relevant compromise, even if this choice, of course, raises the question of the denominator to use to estimate a RR in a context of a vaccination coverage evolving through time. The interest of a sliding window approach was precisely to minimise this issue by considering shorter periods that would still encompass sufficient hospitalisation data for RR estimations. Considering larger periods is mostly an issue when focusing on an ongoing vaccination campaign, which was the case for the boosters but far less for the primary courses during the study period.

## 4. Results and Discussion

Our findings confirm the protection conferred by the vaccines against the risk of hospitalisation in all age groups. RR estimates obtained using a retrospective 4-week sliding window are reported in Figure 1, and estimates obtained for larger periods are reported in Appendix A. As shown in both Figure 1 and Appendix A, our results also confirm notable variations in the apparent protective effect of the vaccine between different age categories, with the RR estimates increasing globally with age [21,22,23]. In particular, sliding window estimates (Figure 1) indicate that at the beginning of the study period, RR estimates started rather high for people above 75 years old (75–85: RR = 0.34, 95% CI = [0.27–0.42]; 85+: RR = 0.65, 95% CI = [0.48–0.87]), which is compatible with an effect of waning immunity, the vaccination campaign for these two age categories having started early 2021. Such waning immunity has also been reported by other studies based on the analysis of surveillance data [24]. Nonetheless, for the same age categories, we then observed a decrease in the RR estimate until the end of 2021 (75–85: RR = 0.20, 95% CI = [0.17–0.23]; 85+: RR = 0.22, 95% CI = [0.18–0.27]), which likely reflects the protective effect of the booster against hospitalisation (as the booster campaign targeted these two oldest age categories). In early January 2022, RR estimates then tended to increase for all age categories, and in particular, for the 85+, followed by the 25–34 age category. While we do not have a specific interpretation for the particular increase in the 25–34 age category, the overall RR increase observed among all age categories from the beginning of January could correspond to an effect of the Omicron (B.1.1.529) variant that had become dominant in Belgium in the same period (Figure 1), and which is known to evade the immunity developed from previous infection/vaccination more efficiently than the Delta variant [11,25,26,27].

Analyses covering larger periods (Appendix A) also displayed important RR variations among age categories, which were consistent with the overall sliding window estimates reported at the end of the same table. Globally, those overall sliding window estimates indicated that the effect of vaccination on the risk of hospitalisation remains high, with RR associated with “at least primary course completed” ranging from 0.10 for the 25–34 to 0.42 for the 85+. Those results also highlight that the RR estimates associated with the “only primary course completed” vaccination scheme (only available for the second half of the study period) approached 0.75 for the 65–74 group and even around 1 for the 75–84 and 85+ groups (75–84: 1.07, 95% CI = [0.95–1.20]; 85+: RR = 0.91, 95% CI = [0.80–1.05]). The latter observations further highlight the importance of the booster shots to maintain a sufficient level of protection against the risk of hospitalisation in these older age categories. Another interesting result is that the RR estimates associated with the “boosted” vs. “only primary course completed” (only available for the second half of the study period) were higher than 0.5 for the <45 years old, which could indicate a lower, yet not negligible, impact of the boosters on the risk of hospitalisation for the youngest age categories. However, it is important to mention that the overall peaks in numbers of hospitalisations have been lower for almost all age groups in the Omicron period compared to the Delta period, as seen in Figure 1B.

In Appendix A, we also report RR values estimated for the overall populations, as well as for all considered age groups combined (25+). Interestingly, those overall estimates tend to be higher than estimates obtained from each age category taken separately. This is an illustration of the Simpson’s Paradox [28,29], a well-known phenomenon in which a trend appears in several groups of data but changes (increases, disappears, or reverses) when the groups are combined. These misleading results can sometimes be obtained from observational data in the presence of confounding factors. In our specific case, we hypothesise this trend is caused by the fact that the confounder (age) is strongly positively associated with the exposure (vaccination) and outcome (severe disease requiring hospitalisation).

According to the ECDC report on vaccine effectiveness for eight European countries, high vaccine effectiveness in preventing severe disease associated with laboratory-confirmed SARS-CoV-2 was registered [30]: the vaccine effectiveness against the risk of COVID-19 hospitalisation ≥14 days after full vaccination (i.e., primary course completed) with any vaccine product was estimated to be 93% during the pre-Delta period and 83% for the Delta variant dominant period. For instance, a study in North Carolina (USA) using a Cox regression model found long-lasting vaccine effectiveness in reducing the risk of hospitalisation for three vaccines for the Alpha variant period, which ranged between 75.2% (95% CI = [72.1–78.0]) and 97.2% (95% CI = [96.1–98.0]) (31). However, as a result of the emergence of the Delta variant, waning immunity was further observed and effectiveness against the risk of hospitalisation decreased by approximately 10–15% [31].

Our RR estimates based on hospital surveillance data are associated with a series of limitations. First, unlike other observational studies [18,32], our estimates were not corrected for confounding factors such as the region of residence, sex, comorbidities, potential exposure differences, previous infection(s), vaccine brand, or time since the last administered vaccine dose. This limitation is inherent to the present national hospitalisation surveillance database used in our study, in which only aggregated data on the vaccination status and the age of the patient, admitted to the hospital for a SARS-CoV-2 infection, have been recorded. Second, from the study start date (06/10/21) to mid-December 2021, recorded vaccination data at the hospital admission did not make the distinction between “primary course completed” and “boosted” vaccination schemes, which prevented us from comparing the associated trends between these schemes during the first half of the study period. Finally, we have to be cautious when analysing RR values estimated for the 75+ group, which represents a relatively small group with specific characteristics such as high vaccine coverage and a higher mortality rate.

While further illustrating the impact of COVID-19 vaccines on the risk of hospitalisation, our results also highlight trends that can be discussed in light of waning immunity, booster campaigns, and the emergence of the Omicron variant. Furthermore, our study also illustrates that aggregated hospital surveillance data on the number of COVID-19 admissions by age and vaccination status can be used in near real-time to obtain and compare trends in vaccine impact on the risk of hospitalisation by age category and vaccination status. Therefore, such data could, for example, serve as early signals for political guidance. Yet, a flexible surveillance system is required to adapt to changing situations (e.g., the introduction of a booster dose). Due to the different limitations outlined above and inherent to the collected data, such analyses should be complemented with prospective cohort studies to account for potential effects of waning vaccine-induced immunity and risk factors at the individual level.

## Figures and Tables

**Figure 1 viruses-14-01315-f001:**
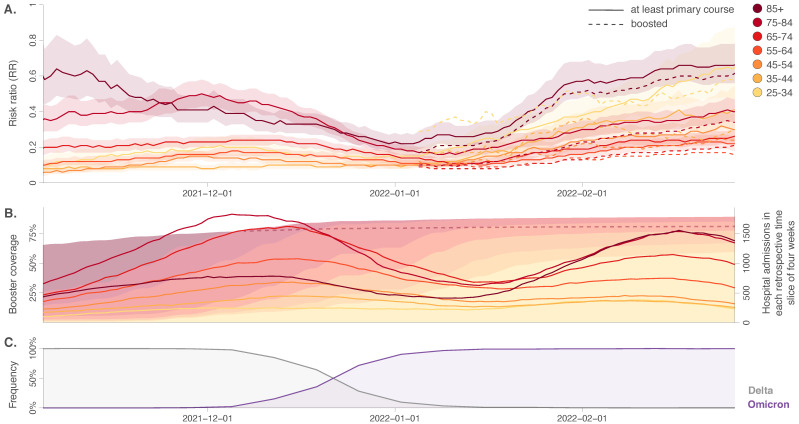
**Approximation of vaccine impact on the risk of hospitalisation through the estimation of risk ratios (RR) between October 2021 and February 2022, based on the analyses of aggregated hospitalisation surveillance data collected in Belgium.** (**A**) We here report RR values estimated for different age categories using a retrospective sliding window of four weeks. RR estimates are associated either with the “at least primary course completed” vaccination scheme (solid curves) or with the “boosted” vaccination scheme (dashed curves). Of note, in Belgium, hospitalised patients having had their booster jab were only registered as “boosted” at their admission from December 8, 2021. Therefore, we are only able to make the distinction between “primary course completed” and “boosted” vaccination schemes for the second half of the study period (15/12/22–28/02/22). Shaded polygons correspond to 95% confidence intervals associated with the “at least primary course completed” vaccination scheme (20). In addition, we also report (**B**) the progression of the Belgian booster vaccination campaign (shaded polygons *) and the evolution of the number of new hospitalisations in each slide of the retrospective sliding window of four weeks for the different age categories (curves), as well as (**C**) the temporal evolution of the relative detection frequency of the two main variants of concern (VOCs) circulating during that period in Belgium. (*) Because hidden on the graph by the progression of the booster vaccination campaign for younger age categories, the related progression for the 85+ is highlighted by a dashed curve from early 2022.

## Data Availability

Publicly available vaccination data were retrieved from the Sciensano website (epistat.wiv-isp.be/COVID, accessed on 10 March 2022) and the number of Belgian people in each age category was obtained from the StatBel database (bestat.statbel.fgov.be, accessed on 10 March 2022).

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
