# Peer review of "Investigating COVID-19 Vaccine Impact on the Risk of Hospitalisation through the Analysis of National Surveillance Data Collected in Belgium"

_viruses, 2022, doi:10.3390/v14061315_

Round 1

Reviewer 1 Report

Thank you for your interesting use of hospital routine surveillance data to assess the impact of the Covid-19 vaccination. As you pointed out, the data and method used have strengths and limitations, but the results provide a logical interpretation of the dynamics of the protection conferred by the vaccines in the Belgian population. The article would benefit from providing more detail on the RR calculation, perhaps with some examples using the actual numbers. To better achieve your goal of "exploring the potential of aggregate surveillance data to monitor potential trends related to the impact of vaccines on the risk of hospitalization," can you highlight some lessons learned from this approach and make recommendations on how to use it. Suggestions on improving the surveillance data and data analysis may be of interest to researchers in other contexts.

Author Response

Thank you for this positive feedback on our work. The lessons learned from the analytical approach is summarised in the last two paragraphs of the Discussion section, but we have now further detailed the RR computation and explicitly mentioned that “a flexible surveillance system is required to adapt to changing situations (e.g. introduction of a booster dose)”.

Reviewer 2 Report

This is an interesting research study about the impact of COVID-19 vaccination on the risk of future hospitalization. The data studied was obtained from hospital surveillance and a publicly available vaccine repository. The authors acknowledge the limitations of the study which include the inability to assess for breakthrough infections and patient origin (geographic region and patient residence - nursing home versus community residing). My suggestions are to proofread for minor language and grammatical corrections. 

Author Response

Thank you for your positive assessment. We have carefully proofread the revised version of our manuscript.